# Government Environmental Regulation and Corporate ESG Performance: Evidence from Natural Resource Accountability Audits in China

**DOI:** 10.3390/ijerph20010447

**Published:** 2022-12-27

**Authors:** Yingzheng Yan, Qiuwang Cheng, Menglan Huang, Qiaohua Lin, Wenhe Lin

**Affiliations:** 1College of Economics and Management, Fujian Agriculture and Forestry University, Fuzhou 350002, China; 2School of Economics and Business Administration, Postdoctoral Research Center of Business Administration, Chongqing University, Chongqing 400044, China

**Keywords:** environmental regulation, ESG indices, sustainable development, government audit

## Abstract

With the increasing global concern for the ecological environment and sustainable development, all countries have proposed environmental regulatory policies to improve the quality of their ecological environments. China has also proposed an environmental regulation policy: Leading an officials’ accountability audit of natural resources (AANR). As the main subject of consuming resources, the sustainability of enterprises has become a focus of all parties. The Environmental, Social, and Governance (ESG) metric measures corporate sustainability. As a result, companies’ ESG performance has gained the community’s attention. Based on data from Chinese A-share listed companies in Shanghai and Shenzhen from 2011 to 2019, this study investigates the role of AANR on the ESG performance of companies via the difference-in-differences (DID) method. This study found that implementing the AANR pilot significantly negatively impacted corporate ESG performance. This result was found to remain robust after passing parallel trend and robustness tests. Further research found that the AANR differed significantly across corporate ownership and regions in corporate ESG performance. First, pilot implementation had a more significant impact on the ESG performance of non-state enterprises. Second, the differences across regions showed that the central region had the most significant impact, followed by the western region, while the eastern region had the most negligible impact. This study will help government departments improve the AANR system and enable companies to focus on their ESG performance.

## 1. Introduction

Recently, the concept of high-quality development characterized by innovation, sustainability, coordination, openness, and sharing has been widely accepted and adopted. This concept promotes the sustainable development of the global economy [1]. In response to this global development trend, China’s economy is gradually shifting from a high-growth stage to a high-quality development stage. Ecological civilization has been incorporated into China’s development requirements, and China has further emphasized implementing the concept of sustainable development. However, China’s economy has been growing at a high rate for a long time. In this process, China has placed excessive emphasis on regional economic construction and development, making economic assessment indicators a vital reference standard in the promotion evaluation system of regional officials. This type of examination requires officials to develop a management model that promotes rapid regional economic development at the expense of the region’s natural resources [2]. However, rapid economic development has seriously damaged natural resources and the ecological environment. Subsequently, China has started to provide vital protection for the construction of ecological civilization by establishing strict systems and enforcing the rule of law at the national level [3]. In this context, The Third Plenary Session of the 18th Central Committee of the Communist Party of China proposed for the first time to conduct an exit AANR for leading cadres and build a lifelong accountability system for damage to the ecological civilization. Since 2014, four pilot batches of AANR have been carried out, and the system was normalized in 2018. Implementing this pilot project can effectively restrain the behavior of regional officials who focus only on the economy but not on resource conservation during their tenure and make local officials pay attention to environmental protection in their jurisdictions [4]. At the same time, as one of the main subjects of environmental pollution and natural resource consumption [5], the introduction of a binding system also constrains the behavior of enterprises in the jurisdiction. Environmental regulations make it necessary for companies to increase their economic value and take more social responsibility regarding resource use and environmental protection in the context of regional development. Therefore, the traditional perception of corporate responsibility is no longer adapted to the requirements of the times for the construction of a global ecological civilization. The concept of responsibility fulfillment is led by the Environmental, Social, and Governance dimensions, which have become the three most important aspects to measure the sustainable development of enterprises. The ESG concept originated from responsible and ethical investments, formally introduced by the United Nations Principles for Responsible Investment in 2006. Since then, the ESG investment concept has started to gain attention. ESG ratings have become a vital indicator for investors to examine whether a company is worth investing in [6]. In 2021, the China Securities Regulatory Commission (CSRC) reformulated and unified the disclosure standards for ESG information in listed companies’ statements, further indicating that the Chinese market is also paying more attention to the ESG information of listed companies.

With the implementation of the AANR pilot, local governments need to intervene and restrain the negative external behavior of enterprises to achieve the desired environmental goals. Therefore, to alleviate the assessment pressure of AANR, local officials will impose stronger environmental regulations on the enterprises in the jurisdictions where they work [7], thus guiding them toward high-quality development. In this context, the present study explores whether corporate ESG performance will be affected by this pilot. Here, we utilize data from Chinese A-share listed companies in Shanghai and Shenzhen from 2011–2019 as the research sample. The AANR pilot is used as a quasi-natural experiment. This article adopts the DID method to explore in depth whether pilot implementation will impact corporate ESG performance. This study offers two main contributions. Most existing studies focus on the impact of corporate ESG performance on corporate value, investments and other factors. However, few studies have examined the impact of regional environmental policies in China on the ESG performance of local firms. This study is the first attempt to include AANR pilot implementation and corporate ESG performance in the same explanatory framework and explore the possible causal relationship between them. The study also finds that the pilot implementation in China is not conducive to enhancing the relationship between the ESG performance of firms in the short run. This effect varies significantly by firm ownership and region. Additionally, the negative impact of the pilot implementation on the ESG performance of non-state enterprises is more significant than that of state-owned enterprises. The pattern of regional differences shows that the central region has the most significant impact, followed by the western region, and the eastern region has the slightest impact.

The rest of the paper is organized as follows. The literature review and hypotheses are described in Section 2. Section 3 describes the article’s research design, including the research methodology, sample selection, and variable definitions. Section 4 presents an analysis of the empirical results. Section 5 provides further analysis. Section 6 provides conclusions and policy recommendations.

## 2. Literature Review and Hypothesis

### 2.1. Literature Review

#### 2.1.1. Research on AANR

Research on AANR has focused on normative and empirical studies. In terms of normative research, the system has clear audit objectives, audit subjects, audit scope, and audit content [8]. The Third Plenary Session of the 18th Party Central Committee proposed to “explore the preparation of natural resources assets and liabilities, the implementation of the leading cadres of natural resources assets out of office audit.” It can be seen that the two are closely related and mutually supportive. As a critical part of AANR, the audit of the natural resources balance sheet also has differences in terms of audit subject, content, and focus. Understanding these differences can help better implement the AANR. However, there are still problems with evaluation under this system, such as the difficulty of coordinating economic responsibility audits with natural resource audits [9]. Therefore, China should promote the construction of an audit evaluation system by actively exploring new models that combine big data technology with audit system implementation science [10]. Based on the above analysis, normative research on auditing outgoing officials’ management of natural resource assets system was developed systematically. For empirical studies, scholars have focused on the impact of the implementation of this pilot system for the fulfillment of corporate environmental responsibility, transformation and upgrading, and the cost of equity capital [7,10,11].

#### 2.1.2. Research on Corporate ESG Performance

High-quality development has become the theme of China’s economic and social development. As the main body of national economic development, whether enterprises can achieve high-quality development plays a decisive role in promoting high-quality development of the national economy. The ESG index evaluation system is highly suitable for the requirements of high-quality development and has attracted widespread attention from scholars for this reason. Most studies on corporate ESG performance have highlighted the impact of corporate ESG performance on corporate performance, investment, and value. A company’s ESG performance can influence its corporate value [12,13,14,15,16], and good ESG performance and high-quality ESG disclosure can help improve corporate financial performance [17,18,19,20]. Corporate ESG performance also impacts corporate financialization, financing difficulty, and corporate investment efficiency [21,22,23]. In addition, a few scholars focused on the factors influencing the ESG performance of firms. These researchers argue that local government debt, the characteristics of the market in which the firm is located, the capital market’s openness, and the managers’ education levels have an impact on corporate ESG performance [24,25,26,27].

#### 2.1.3. Research on the Impact of AANR on Corporate ESG Performance

The AANR has forced regional officials to strengthen monitoring of the pollution status of enterprises in their jurisdictions, prompting them to focus on their own environmental and social responsibilities. The AANR system can promote corporate environmental responsibility [11] and social responsibility [28], both of which are indicators that companies need to refer to when rating their ESG performance. Consequently, implementing AANR will impact ESG. Thus, how does the implementation of AANR affect the ESG performance of companies? This question is the central question of this paper. In the AANR Regulations (Trial) promulgated in 2017, it is stipulated that the subjects of AARN should be responsible for protecting the natural resource assets and ecological environment. Based on this background, it is clear that the AARN system is an environmental regulation policy. Therefore, implementing the AANR will prompt regional leaders to impose environmental regulations on companies in their jurisdictions. However, it remains controversial whether environmental regulations can positively affect business development. On the one hand, some scholars argue that environmental regulation can significantly improve firm performance [29] and promote technological innovation transformation [30], satisfying the “Porter hypothesis” theory. On the other hand, some scholars argue that environmental regulation increases the financial risk of firms and makes them less financially leveraged. This perspective is inconsistent with Porter’s hypothesis that environmental regulations help guide firms increase their financial leverage and actively explore capital markets [31].

In summary, research on AANR and corporate ESG performance has focused on the economic effects of pilot implementation and the role of corporate ESG performance. No studies have yet included AANR and corporate ESG performance in the same analytical framework. In sporadic studies, some scholars have argued that AANR can promote the fulfillment of corporate environmental or social responsibility [11,28]. However, AANR is also limited to analyzing the policy effects of the pilot from a single responsibility perspective, failing to consider the performance of all aspects of ESG in an integrated manner.

### 2.2. Hypothesis

The implementation of AANR has prompted companies to pay more attention to their environmental and social responsibilities. However, companies may not be able to perform at a good level under high pressure environmental regulations. Under the implementation of this system, companies will be more inclined to improve their environmental responsibilities [11], making it costly for them to invest in environmental management [32], which may, in turn, reduce the level of corporate social responsibility implementation and internal corporate governance, negatively affecting overall ESG performance.

From the perspective of corporate social responsibility implementation, based on the theory of external information transfer, the “Regulations on the Leading officials’ accountability audit of natural resources (for Trial Implementation)” promulgated in 2017 [33] clearly stipulates that leading cadres should fulfill their responsibilities for environmental protection and improvement during their tenure. As one of the most important subjects of environmental pollution and resource consumption, companies will receive more attention from regional regulators, leading to a rise in corporate risk and sending negative signals to the market. To mitigate the negative impact of environmental regulations on companies, companies may reduce their impacts directly by focusing on their environmental responsibilities in order to address the concerns of investors and attract them [34]. In this scenario, the role of corporate social responsibility is the same. Empirically, many companies take socially responsible measures to maintain a good corporate image and attract investors [35]. In a study on the impact of consumer satisfaction and loyalty in the private banking sector in Peru, scholars found that corporate social responsibility has a positive impact on customer loyalty. The fulfilment of social responsibility contributed to creating preference and loyalty for the company [28]. Improving the fulfilment of corporate environmental and social responsibility can attract investors and increase customer loyalty. However, companies have limited resources, and fulfilling these two responsibilities requires organizations to make significant financial investments [32,36], which may hinder profitability. In 2007, the adoption of Indonesian Law No. 40 sparked a major debate on the nature of corporate social responsibility. The study found that the implementation of both environmental and social responsibility increases the cost of operations and consumes corporate resources. The resulting trade-off is that it is easy for companies to focus on their own environmental responsibilities at the expense of their social responsibilities.

In terms of the level of internal corporate governance of the business. Firstly, based on the principal-agent theory, there is an incentive for corporate managers to invest inefficiently and pursue private gain. There is also an incentive for majority shareholders to usurp the interests of minority shareholders. When the negative impact of the above-mentioned actions on the company have the opportunity to be hidden, the incentive increases for corporate managers and major shareholders to pursue their private interests. Management and major shareholders may exaggerate the negative impacts of the implementation of AANR on company operations. The aim of this exaggeration is to hide ineffective investments and emptying [37,38] and to attribute the negative impacts of these actions on the company’s operations to AANR, leading to a decline in the level of internal corporate governance. Secondly, based on neoclassical economic theory, the production behavior of enterprises may lead to negative externality problems, which require effective regulation by the government when the market is unable to regulate the adverse consequences of negative externalities on social development. It is known that a company’s focus on fulfilling its environmental responsibilities leads to unproductive cost investments in environmental management. Subsequently, the large compliance costs incurred by the policy will crowd out productive expenditures [39], breaking down the costs of capital structure, making internal governance more difficult, and increasing operational risk. Some scholars have studied relevant industries in Europe and the North through earnings and cash-flow forecasts and found that some companies with good environmental performance have lower stock returns. This result further shows that companies that have implemented environmental initiatives and are focused on environmental protection have a lower level of internal corporate governance and are less able to cope with external shocks [40]. Other researchers have found that good environmental performance in coal companies reduces the level of governance within coal companies, resulting in lower corporate earnings [41].

Overall, due to the limited governance efforts and resources of a company, it may not be possible to take into account the overall development of corporate ESG. Therefore, to a certain extent, the implementation of AANR will increase the commitments of enterprises to their own environmental responsibilities while neglecting their fulfilment. Overall, due to the limited governance efforts and resources of a company, it may not be possible to take into account the overall development of corporate ESG. Therefore, to a certain extent, the implementation of AANR will increase the commitment of enterprises to their environmental responsibilities while neglecting the fulfilment of corporate social responsibility and reducing the level of the internal corporate governance of enterprises. Therefore, the following hypothesis is proposed in this study:

**H1:** 
*Auditing outgoing officials’ management of natural-resource-asset pilot implementation hurts corporate ESG performance.*


## 3. Study Design

### 3.1. Sample Selection and Data Sources

Pilot audits of AANR began in 2014. Moreover, the number of pilots increased continuously from 2015–2017. This study analyzed the impact of pilot implementation on corporate ESG performance using listed companies in the pilot region in 2014 as the treatment group and listed companies in the non-pilot region as the control group. We learned about the first batch of pilot cities in 2014 by exploring the China Audit Yearbook and the official websites of the audit offices of provinces, cities, and autonomous regions, including Beijing (Fengtai District, Yanqing District), Yunnan Province (Dali Bai Autonomous Prefecture), Shaanxi Province (Xi’an City), Fujian Province (Fuzhou City, Wuyishan City), Guangxi Zhuang Autonomous Region, Jiangsu Province (Lianyungang City, Nantong City), Guizhou Province (Chishui City, Qiannan Buyi Miao Autonomous Prefecture), Guangdong Province (Shenzhen City), Hunan Province (Loudi City), Inner Mongolia Autonomous Region (Chifeng City, Erdos City, Xing’an League), Hubei Province (Huanggang City, Wuhan City), and Shandong Province (Qingdao City, Yantai City).

Considering the global financial crisis from 2007–2009 and the impact generated by COVID-19 in 2020, we selected 2011–2019 as the data interval. We chose Shanghai and Shenzhen A-share listed companies as the research sample and processed the sample according to the following steps: (1) Companies without ESG ratings were excluded from this study based on the 2011 ESG ratings of the Sino-Securities Index. These companies all went public after 2011, so no 2011 ESG ratings were available for the Sino-Securities Index. (2) We excluded ST and *ST category companies and (3) listed companies in regions where the pilot began in 2015–2017 to reduce the impact of subsequent piloting efforts. (4) This paper excluded companies with missing or abnormal data for key variables and obtained 8892 observations from 988 listed companies from 2011 to 2019. We obtained information on the audit pilot areas through the China Audit Yearbook and the websites of provincial audit offices and municipal audit bureaus. ESG ratings and other control variables were obtained from the WIND database. Moreover, the local environmental regulation data were taken from the China Statistical Yearbook and China Environmental Statistical Yearbook. Data on provincial and regional innovation levels were obtained from the “China Regional Innovation Capacity Evaluation Report 2011–2019”. The China Association for Strategic Research in Science and Technology Development prepared the report in cooperation with the China Innovation and Entrepreneurship Management Research Center of the University of Chinese Academy of Sciences. The article uses the Stata version 14.0 software. All the variables are in logarithmic form to reduce the influence of extreme outliers on the regression results. Continuous variables with “outliers” were first Winsorized, i.e., at the 2.5% and 97.5% percentile, respectively, before being transformed into logarithmic form.

### 3.2. Variable Selection and Descriptive Statistics Results

The ESG rating system in China is still in its initial stage. The ESG information disclosure for different listed companies in China has different indicators and different calibers, and the data cannot be compared, which makes it infeasible to construct standalone ESG indicators. Currently recognized third-party ESG rating systems include SynTao Green Finance (SGF) and Sino-Securities Index, China Alliance of Social Value Investment (CASVI). Compared to the Sino-Securities ESG Index, other ESG evaluation systems all have a certain degree of narrow coverage and difficulty in obtaining data. For example, CASVI and SynTao Green Finance only cover a portion of the constituents and are not yet live in databases such as WIND and CSMAR. The Sino-Securities ESG Index refers to the mainstream ESG rating framework in foreign countries and adds indicators with Chinese characteristics, subdividing the three pillars of environment, society, and governance into 14 themes and 26 key indicators (See Table 1), with the advantages of adaptation to the Chinese market, wide coverage, and very timely data. To adapt the ESG rating framework to the Chinese market and Chinese companies, the Sino-Securities ESG Index was chosen to represent the ESG performance of companies with reference to the relevant literature [42,43,44]. The explanatory variable in this study is corporate ESG performance (lnESG), which is represented by the Sino-Securities ESG Index. The dependent variable in this study is corporate ESG performance (lnESG), which is expressed using the Sino-Securities ESG Index. The Sino-Securities ESG Index is divided into nine levels, from excellent to poor, including AAA-C. The higher the score, the better the ESG performance of the company. We manually assigned “AAA-C” to “9-1”.

The core explanatory variable in this study is the interaction term between auditing outgoing officials’ management of natural-resource-asset pilots and time (TREAT*POST), which represents whether or not the audit pilot was implemented, setting listed companies in pilot cities as the treatment group and listed companies in non-pilot cities as the control group.

In order to control variables that may affect the ESG performance of firms as much as possible, we refer to relevant studies [42,45,46] and set the following control variables: (1) the accounts receivable turnover ratio (lnART), measured using the primary business’s revenue ratio to average accounts receivable; (2) the top 10 shareholders’ shareholding ratio (lnTOPIC10), measured using the top 10 shareholders’ shareholdings ratio to the total number of shares; (3) return on assets (lnROA), measured using the ratio of a firm’s net interest rate to its total assets; (4) Loan of Asset Ratio (lnLEV), measured using the ratio of total liabilities at the end of the period to total assets at the end of the period; (5) firm size (lnSIZE), expressed using the natural logarithm of total assets; and (6) regional innovation level (lnINNOV), measured using the total utility value of China’s regional innovation capacity from the China Regional Innovation Capacity Evaluation Report. The explanations of variables and data sources are shown in Table 2, and the results of descriptive statistics for each variable are shown in Table 3.

## 4. Analysis of Empirical Results

### 4.1. Results of Baseline Regression Analysis

Before the baseline regression analysis, it is first necessary to test the variables for the presence of multicollinearity problems. The test criterion for multicollinearity is usually a variance inflation factor VIF value less than 10. Subsequently, it is necessary to select the appropriate model from among the mixed Ordinary Least Square (OLS), random, and fixed effects models for regression analysis. After comparing the mixed OLS model with the fixed effects model, the F-test value of 6.69 with a significance level of 0.000 indicated that the fixed effects model should be selected. For the choice of the fixed effects model and random effects model, the Hausman test results showed that the χ2(8) value is 83.80 with a significance level of 0.000, which indicates that the fixed effects model is better than the random effects model. Therefore, we used the fixed-effects model as the baseline regression model in this study. The empirical analysis was conducted using the DID method, and the results are shown in Table 4. From column (3), it is clear that TREAT × POST is significant and has a negative coefficient at the 1% level when no control variables are considered. This result indicates that auditing outgoing officials’ management of natural resource assets reduces corporate ESG performance. From column (4), it is clear that TREAT × POST is significant at a 1% level with a negative coefficient after considering the control variables. This result also indicates that the implementation of auditing outgoing officials’ management of natural-resource-asset pilot decreases the ESG performance of firms, verifying the validity of the hypothesis. In terms of other control variables, Top 10 shareholders’ share-holding ratio (lnTOPIC10), Return on assets (lnROA), Accounts receivable turnover ratio (lnART), firm size (lnSIZE), and regional innovation level (lnINNOV) have a significant positive effect on firms’ ESG performance. This result shows that the higher the top 10 shareholders’ Accounts receivable turnover ratio, the larger the firm size; additionally, the higher the regional innovation level, the better the ESG performance of the firm. Loan of Asset Ratio (lnLEV) has a significant negative effect on ESG performance, indicating that as the Loan of Asset Ratio increases, it will significantly reduce the ESG performance of the firm.

The reasons for such a positive impact are as follows. Firstly, it is clear from resource dependence theory that, when the board of directors is larger, the firm has more resources and is subject to increased scrutiny, thus enabling improved decision-making, better corporate performance, and improved ESG performance. Therefore, the higher the top 10 shareholders’ shareholding ratio, the better the ESG performance of the firm. Secondly, good corporate profitability and operational capabilities are a reflection of a firm’s level of corporate governance. These two capabilities are measured by the return on assets and accounts receivable turnover ratio. Therefore, the greater the return on assets and the higher the accounts receivable turnover ratio, the better the ESG performance of the firm [47]. Thirdly, a larger firm size implies that the firm has more resources, and firm size can have a positive impact on a firm’s ESG score, which is consistent with past research [48]. Fourthly, at higher regional innovation levels, firms are better able to cope with shocks caused by environmental regulation [49]. Thus, the regional innovation level shows a positive relationship with corporate ESG performance. In addition, the loan of asset ratio has a significant negative impact on corporate ESG performance, indicating that as the loan of asset ratio increases, corporate business risk increases and will significantly reduce corporate ESG performance, similar to the findings of other researchers [46].

### 4.2. Parallel Trend Test

After referring to the research methods of other researchers [11,50,51], we decided to use the DID method for the study. However, the parallel trend assumption is a prerequisite for the use of DID in empirical papers. DID can be used only if the target variables in the treatment and control groups satisfy the parallel trend assumption before policy is implemented. Conversely, if there is some ex ante difference between the treatment and control groups, then it represents a high probability that there are other factors influencing the variation in the explained variables, which is when the triple difference method (DDD) is used. The results of the parallel trend test in this paper satisfied the condition that the DID method can be used. In addition, we found that DID models have been widely used in many areas to test the impact and effectiveness of policy reforms. Examples include health care bill evaluations, disability employment system evaluations, welfare reform evaluations, and pre-kindergarten program evaluations [52,53,54,55]. The following section elaborates on our parallel test results.

Based on the results of the benchmark regression, implementing the AANR reduced corporate ESG performance. However, whether the baseline regression results can genuinely reflect the impact of auditing outgoing officials’ management of natural resource assets on corporate ESG performance needs to be tested for parallel trends. For the empirical evidence, the following dummy variables are set: three periods, two periods, and one period before pilot implementation (pre3, pre2, and pre1, respectively); the current period of pilot implementation (current); and the first, second, third, fourth, and fifth periods after pilot implementation (post1, post2, post3, post4, and post5, respectively). Using the model shown in column (4) of the baseline regression in Table 4, we tested the impacts and results of the pilot and firm ESG performance based on parallel trends. Figure 1 demonstrates the results. In the parallel trend test, referring to existing studies [56], the second year before pilot implementation (i.e., 2012) was used as the baseline group in the test, and the condition for judging the validity of the parallel trend test was that neither pre3 nor pre1 was significant. As shown in Figure 1, the confidence interval of the regression coefficients for AANR before pilot implementation contained 0 when controlling for other variables. This result shows that the trend of changes in the experimental and control groups remained unchanged. Thus, this study passed the parallel trend test and excluded the endogeneity problem. The passage of this test also verifies the robustness of the benchmark regression. In addition, as seen in Figure 1, the pilot significantly impacted corporate ESG performance in the year of implementation, and this impact continued through 2019.

### 4.3. Robustness Tests

The baseline regression results indicate that implementation of the pilot reduced corporate ESG performance. We used sample exclusion and core explanatory variable replacement to test the robustness of this result.

#### 4.3.1. Sample Exclusion Method

Implementing the AANR pilot may have significantly impacted resource-based and heavily polluting firms. However, samples from other industries were not excluded from the baseline regressions, which may lead to inconsistent regression results. To this end, we investigated whether AANR also impacted corporate ESG performance, using only a sample of resource-based and heavily polluting firms. Columns (5) and (6) in Table 5 show the results.

As seen from the table, the baseline regression results hold after the study excludes the samples of other companies. This result suggests that the baseline regression results are relatively robust.

#### 4.3.2. Core Explanatory Variable Substitution Method

This study uses the AANR pilot as a quasi-natural experiment to explore the impact on corporate ESG performance. The pilot is an environmental regulation [57] that will significantly impact regional environmental governance. To verify the robustness of the baseline regression results, an attempt was made to use the regional level of environmental regulation (provincial) as a proxy variable for this pilot. To explore whether regional environmental regulation affects firms’ ESG performance, the level of regional environmental regulation (lnER) was measured based on existing practices [58]. The entropy method was utilized. Regional industrial wastewater emissions, industrial SO2 emissions, and industrial smoke emissions were used for calculations. Additionally, the test was conducted using a sample of resource-based and heavily polluting firms. The effects of regional environmental regulation levels on the ESG performance of firms are shown in columns (7) and (8) of Table 5. As seen from the table, replacing the core explanatory variables with the level of regional environmental regulation still has a significant adverse effect on the ESG performance of firms.

## 5. Further Analysis

Implementation of AANR pilot reduced corporate ESG performance, as shown by the Differences-in-Differences method estimation results. However, the reasons for the large differences in the ESG performance of companies include the large differences between different regions, as well as the large differences in ownership of different companies. Therefore, we further explored whether there are significant differences in the effects of AANR points on corporate ESG performance from two perspectives: different regions and different types of corporate property rights.

### 5.1. The Impact of Piloting on Corporate ESG Performance Based on Regional Differences

China is a vast country with significant geographical variability in the level of economic development in different regions. Different levels of economic development make it possible for the implementation of the pilot to show differences in corporate ESG performance. Our study divides China’s 31 provinces into eastern, central, and western regions. The results are shown in Table 6. As shown in this table, AANR pilots significantly negatively impacted corporate ESG performance in all three regions. However, there were differences in the magnitude of the impact. The central region presented the most significant impacts, followed by the western region, while the eastern region had the slightest impact. The eastern region had the most negligible impact because economic development in the eastern region is shifting from being capital-factor driven to innovation-driven, and the pace of the shift is significantly faster than that in the western and eastern regions. Additionally, the industrial structure in the eastern region is being optimized, and a new industrial structure is being built, with science and technology manufacturing and production services as the core industries. Moreover, the building speed is significantly faster than that in the western and eastern regions [59]. It can be seen that the economic development model and industrial structure of the eastern region are more conducive for enterprises to cope with upgrading and transformation in the development process. Therefore, the eastern region is better able to cope with the rising cost pressures introduced by environmental regulations [32] and mitigate the inhibiting effects of the exit audit system on corporate ESG performance. The central and western regions have lower labor costs, richer natural resources, and more government incentives for development than the eastern regions. These advantages have led to more pollution-intensive enterprises in the central and western regions. Moreover, the number of such enterprises is significantly higher in the central region than in the western region [60]. The implementation of environmental regulations has led regional leaders to strengthen the environmental management of intensive enterprises in their jurisdictions [4]. However, the central region’s economic development and industrial structure are not as well established as those of the eastern region. Companies in the central region are unable to upgrade and transform quickly to cope with the higher intensity of environmental regulations. Therefore, the exit audit system may have a more significant inhibiting effect on the ESG performance of enterprises in the central region.

### 5.2. The Impact of Piloting on Corporate ESG Performance Based on Differences in Corporate Ownership

Pilot auditing of officials’ management of natural resource assets is an essential institutional innovation. Implementing this system will have different degrees of impact on enterprises with different property rights in the pilot area. This study examined the impact of the pilot exercise on ESG performance under different corporate ownership types (state-owned and non-state-owned enterprises). The regression results are shown in Table 7. As shown in this table, AANR has a more significant adverse effect on non-state enterprises. In contrast, the effect on state-owned enterprises is insignificant. This low impact is because state-owned enterprises are better able to cope with external risks than non-state-owned enterprises and have institutional, human, and resource advantages [61]. In addition, state-owned enterprises have not yet demonstrated the advantages of innovation and marketability [62]. However, rich innovation resources are possessed by state-owned enterprises. Abundant resources make state-owned enterprises better equipped to handle the external risks associated with environmental governance. Therefore, AANR was found to significantly negatively impact the ESG performance of non-state enterprises. In contrast, the impact on state-owned enterprises was not significant.

## 6. Conclusions and Policy Recommendations

Globally, many countries such as China are pursuing rapid economic development at the expense of environmental pollution and sacrificing the ecological environment. In order to follow the global concept of sustainable development, China’s economy has started to shift from high growth to a stage of high-quality development. Enterprises are an essential part of high economic development. Guiding the high-quality development of enterprises has become the focus of China’s policy implementation. Therefore, it is of great theoretical and practical importance to explore the policy implications of AANR on corporate ESG performance. This study selected listed companies in Shanghai and Shenzhen A-shares from 2011–2019 as its research sample. The DID method was used to analyze the impacts of the pilot AANR on corporate ESG performance. We found that implementation of the AANR pilot had a significant negative impact on corporate ESG performance, i.e., the implementation of the auditing pilot was not conducive to improving corporate ESG performance. The robustness of this finding was further demonstrated by the parallel trend test and robustness test. This result was found to be robust through a robustness test. The heterogeneity analysis found that the impact of audit pilots on corporate ESG can vary significantly across corporate ownership and regions. The pilot had a more significant negative impact on the ESG performance of non-state than state-owned enterprises. Regionally, the results showed a pattern of differences, with the most significant impacts observed in the central region, followed by the western region. The most negligible impact was observed in the eastern region. The results of our research could convince the government to improve the system of auditing outgoing officials’ management of natural resource assets and encourage companies to implement ESG concepts. Based on the above findings, the following policy insights were obtained.
(1)The government should analyze the policy impacts of AANR implementation on micro enterprises and feed the results back to the relevant government departments. In the process of promoting the implementation of the system, the government should strengthen support for enterprises. This measure could mitigate the decline in corporate ESG performance produced by factors such as higher environmental management costs.(2)Relevant government departments need to improve their enforcement of AANR and further implement and improve AANR. The system should also be rationalized based on differences in property rights. The government should strengthen its support for non-state enterprises to reduce the negative impacts of the implementation of the system on the ESG of relevant enterprises and narrow the ESG performance gap between state-owned and non-state-owned enterprises in the implementation of the exit audit system. This measures would help obviate the disadvantages of AANR.(3)Environmental management costs vary from region to region. Therefore, there is a need for differentiated AANR standards in different regions. Companies in regions with high economic levels, rapid technological development, and developed support systems in China have a strong corporate capacity to cope with changes in the external environment. Over time, these areas could raise the standard of environmental regulation. Conversely, for regions with low economic levels, slow development of science and technology, and less-developed support systems, the standard could be lowered accordingly, enabling progress to be gradual and progressive. In this way, the differences in ESG performance between Eastern and Midwestern companies in the implementation of AANR could be reduced.

In addition, we have the following shortcomings and outlook. On the one hand, in this paper, we analyzed the policy effects of AANR from the perspective of overall corporate ESG performance. In the future, studying the effects of AANR on E, S, and G indicators will be one of our research directions. On the other hand, in future studies, we will consider lengthening the time period of the study and exploring the topic in more depth.

## Figures and Tables

**Figure 1 ijerph-20-00447-f001:**
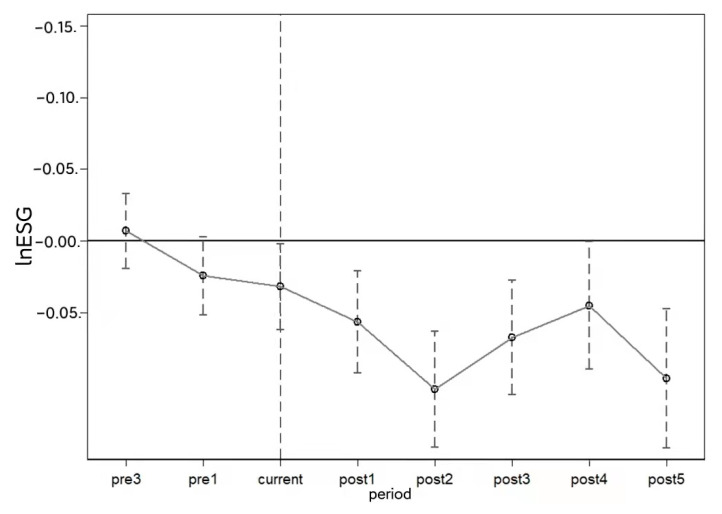
Parallel trend test: logarithm of corporate ESG performance.

**Table 1 ijerph-20-00447-t001:** Composition of Sino-Securities ESG Index specific rating indicators.

Three Aspects	14 Themes	26 Key Indicators
Environment	Environmental Management System	Environmental Management System
	Green Management Objectives	A low-carbon plan or goal
		Green Management Plan
	Green Products	Carbon Footprint
		Sustainable products or services
	External Environmental Certification	Product or company receives environmental certification
	Environmental violations	Environmental violations
Social	Institutional System	Social Responsibility Report
	Health and Safety	Goals or plans to reduce safety incidents
		Negative business events
		Trends in operating accidents
	Social Contribution	Social Responsibility Related Donations
		Employee growth rate
		Countryside Revitalization
	Quality Management	Product or company receives quality certification
Governance	System Building	Corporate self-ESG monitoring
	Governance Structure	Connected Transactions
		Board Independence
	Operating Risk	Overall financial credibility
		Asset Quality
		Short-term debt service risk
		Equity Pledge Risk
		Quality of Information Disclosure
	External Discipline	Violations of laws and regulations by listed companies and subsidiaries
		Violations by executives and shareholders
	Business Activities	Tax Transparency

Note: Data from Sino-Securities ESG Index public official data (https://www.chindices.com/esg-ratings.html#esg-ratings-methodology (accessed on 21 October 2022).

**Table 2 ijerph-20-00447-t002:** Variable explanation and source.

Variables	Variables Explanation	Data Sources
Corporate ESG Performance	ESG performance is categorized into “C-AAA” grades with a score of “1–9”, respectively	Data from the Sino-Securities ESG Index scoring system in the WIND database
Audit Pilot	If the company is registered in the pilot area, take 1, otherwise take 0; If the company is established after the start of the pilot take 1, otherwise take 0; The final variable value is the product of these two items	Pilot information from the “China Audit Yearbook” and the official website of the provincial and municipal audit bureaus; Company information from WIND database
Top 10 shareholders’ shareholding ratio	The top 10 shareholders’ shareholdings ratio to the total number of shares	Data from WIND database
Return on assets	The ratio of a firm’s net interest rate to its total assets	Data from WIND database
Loan of Asset Ratio	Ratio of total liabilities at the end of the period to total assets at the end of the period	Data from WIND database
Accounts receivable turnover ratio	Primary business’s revenue ratio to average accounts receivable	Data from WIND database
Firm size	The natural logarithm of total assets	Data from WIND database
regional innovation level	Total utility value of China’s regional innovation capacity	Data from comes the China Regional Innovation Capacity Evaluation Report

**Table 3 ijerph-20-00447-t003:** Results of descriptive statistics of variables.

Variables	Symbols	Observations	Average	Standard Deviation	Min.	Max.
Corporate ESG Performance	lnESG	8892	1.325	0.350	0.000	2.079
Audit Pilot	TREAT*POST	8892	0.212	0.409	0	1
Top 10 shareholders’ shareholding ratio	lnTOPIC10	8892	0.439	0.100	0.086	0.669
Return on assets	lnROA	8892	0.711	0.027	0.626	0.776
Loan of Asset Ratio	lnLEV	8892	0.353	0.145	0.071	0.623
Accounts receivable turnover ratio	lnART	8892	2.319	1.225	0.747	5.991
Firm size	lnSIZE	8892	12.940	1.285	5.731	18.480
regional innovation level	lnINNOV	8892	3.565	0.364	2.759	4.087

**Table 4 ijerph-20-00447-t004:** Baseline regression results.

Average	(1) Mixed OLS	(2) RE	(3) FE	(4) FE
TREAT × POST	−0.056 *** (−5.88)	−0.066 *** (−4.66)	−0.055 *** (−3.77)	−0.058 *** (−3.73)
lnTOPIC10	0.023 (0.64)	0.131 ** (2.19)	—	0.187 ** (2.37)
lnROA	1.448 *** (8.51)	0.853 *** (4.52)	—	0.643 *** (3.25)
lnLEV	−0.422 *** (−13.51)	−0.344 *** (−7.06)	—	−0.306 *** (−4.98)
lnART	0.005 (1.63)	0.015 *** (3.17)	—	0.021 *** (2.71)
lnSIZE	0.079 *** (24.56)	0.053 *** (8.34)	—	0.033 *** (3.46)
lnINNOV	0.061 *** (5.54)	0.082 *** (3.66)	—	0.144 ** (2.44)
Constant term	−0.803 *** (−6.51)	−0.222 (−1.27)	1.337 *** (432.60)	−0.081 (−0.28)
R-squared	0.117	—	0.004	0.027
Corporate fixed effects	—	—	YES	YES
Observations	8892	8892	8892	8892
Number of companies	988	988	988	988

Note: The t-statistic is in parentheses; *** *p* < 0.01, ** *p* < 0.05; Standard error using robust standard error.

**Table 5 ijerph-20-00447-t005:** Robustness test results.

Variables	(5) FE	(6) FE	(7) FE	(8) FE
TREAT × POST	−0.073 *** (−2.67)	−0.077 *** (−2.62)	—	—
lnER	—	—	−0.110 * (−1.82)	−0.111 * (−1.89)
Constant term	1.320 *** (353.10)	0.165 (0.34)	1.367 *** (43.85)	0.257 (0.53)
Control variables	NO	YES	NO	YES
R-squared	0.004	0.018	0.001	0.016
Corporate fixed effects	YES	YES	YES	YES
Observations	4237	4237	4203	4203
Number of companies	524	524	519	519

Note: The t-statistic is in parentheses; *** *p* < 0.01, * *p* < 0.1; standard error using robust standard error.

**Table 6 ijerph-20-00447-t006:** Results of regional heterogeneity test.

Variables	(9) Eastern Region	(10) Central Region	(11) Western Region
TREAT × POST	−0.042 ** (−2.47)	−0.122 *** (−2.98)	−0.084 * (−1.90)
Constant term	−0.118 (−0.30)	−0.965 * (−1.70)	0.772 (1.39)
Control variables	YES	YES	YES
R-squared	0.030	0.039	0.025
Corporate fixed effects	YES	YES	YES
Observations	5373	1854	1665
Number of companies	597	206	185

Note: (1) The t-statistic is in parentheses; *** *p* < 0.01, ** *p* < 0.05, * *p* < 0.1; standard error using robust standard error. (2) In the regional division, the eastern region includes 11 provinces (or municipalities directly under the Central Government): Beijing, Tianjin, Hebei, Liaoning, Shanghai, Jiangsu, Zhejiang, Fujian, Shandong, Guangdong, and Hainan. The central region includes eight provinces: Shanxi, Jilin, Heilongjiang, Anhui, Jiangxi, Henan, Hubei, and Hunan. The western region includes 12 provinces (or autonomous regions and municipalities directly under the Central Government): Sichuan, Guizhou, Yunnan Shaanxi, Gansu, Qinghai, Chongqing, Inner Mongolia, Guangxi, Tibet, Ningxia, and Xinjiang.

**Table 7 ijerph-20-00447-t007:** Results of the test for heterogeneity of corporate property rights.

Variables	(12) Non-State Enterprises	(13) State-Owned Enterprises
TREAT × POST	−0.073 *** (−3.63)	−0.022 (−0.98)
Constant term	−0.301 (−0.74)	0.028 (0.07)
Control variables	YES	YES
R-squared	0.038	0.020
Observations	5310	3582
Number of companies	627	445

Note: The t-statistic is in parentheses; *** *p* < 0.01; standard error using robust standard error.

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
