# Peer review of "Government Environmental Regulation and Corporate ESG Performance: Evidence from Natural Resource Accountability Audits in China"

_ijerph, 2022, doi:10.3390/ijerph20010447_

Round 1

Reviewer 1 Report

The format of this paper is relatively standard and logical, but the research topic is slightly different from green production. And there are three problems: first, there is no explanation of why to choose those cities in those provinces; Second, too many Chinese documents are cited in the references; Third, the selection of indicators is lack of standardization, which requires reference to indicators in international journals.

Author Response

请参考附件

Reviewer 2 Report

Although the authors try to elaborate the contribution of the research of the paper relative to Zeng et al.(2022). But unfortunately, I got less relevant information from the paper. For example, the hypothesis section of the paper still emphasizes environmental issues. How does the integrated manner show?

Although the authors mention "in the Porter hypothesis, the innovation compensation effect cannot fully compensate for the cost of corporate environmental management", the Porter hypothesis is not always false. If Porter hypothesis holds, will AANR lead to the improvement of ESG?

I would also like to know empirically why AANR hurts ESG performance. This conclusion is significantly different from Zeng et al.(2022).

Reviewer 3 Report

Based on Chinese A-share companies in Shanghai and Shenzhen from 2011 to 2019, this study explores the role of AANR on companies' ESG performance using the difference in differences (DID) method. The study concludes that the implementation of the AANR pilot significantly negatively impacted corporate ESG performance. This conclusion remains robust after the authors pass parallel trend and robustness tests.

I think the subject of the study is current and interesting. The article is well written overall, but I think some parts need improvement. My opinions and suggestions regarding the article are as follows:

First of all, there are some typos in the article. It would be appropriate to check the English of the article by a native speaker.

In the article, “The explanatory variable in this study is corporate ESG performance (lnESG), which 233 is expressed using the ESG rating of Huazheng.” statement is included. In the article, ESG performance was used as a dependent variable in the models, but was written as an explanatory variable in the text. The wrong part needs to be corrected.

It would be appropriate to include a table with the explanations and sources of the variables used in the study. Not enough information has been given about the part of how ESG performance is determined.

No information was given about the method used in the study and why this method was preferred. The methodology part needs to be written more clearly.

The DID abbreviation is used for both the Audit Pilot variable and the difference-in-differences (DID) method. Confusion arises for the reader. It would be appropriate to use different abbreviations.

If control variables are used in the models, it would be appropriate to indicate YES, otherwise NO. It is often used in this way in the literature.

The coefficients obtained in econometric analyzes need to be interpreted. It would be appropriate to explain why this effect is so. In addition, studies that find similar or different results in the literature, if any, should be presented to the reader and the results should be interpreted comparatively. In addition, the coefficient of the Audit Pilot variable was found to be negative in the study. Contrary to expectations, how do the authors explain the negative result of this coefficient? I couldn't find any explanation about this in the text. It would be more appropriate for the authors to focus on the interpretation of these signs.

In the conclusion part, it would be more appropriate to emphasize the results obtained from econometric analyzes. The policy recommendations given in this section should be consistent with the findings of the study. What suggestions do the authors offer if the effect of the pilot application was found to be negative?

Reviewer 4 Report

 This is an interesting question. However, this article needs significant changes.

AANR is a policy for local environmental protection, but ESG covers a wide range of objects, which may affect the basis of inference. It is recommended to make some robustness for environmental indicators alone. Correspondingly, some documents on environmental information disclosure are missing.

The research design of this paper may need to be improved to make the conclusion more credible. The policies in this paper are implemented gradually. It may be inappropriate to use conventional DID strategies. Overlapping DIDs or multi-time point DIDs may be considered. It can be seen from the Parallel trend test that this difference already existed before the policy, possibly because the time before the policy occurred was too short to be noticeable.

Concerning similar literature in JFE, JE, JAE, etc., this article can start from 2007. On the one hand, it expands the length of time before the policy; on the other hand, the impact of the financial crisis on China is smaller than expected.

It may be unconvincing to delete enterprises that do not disclose ESG scores. Many versions of ESG scores of Chinese listed companies refer to existing literature, but few articles use huazheng ESG alone. This is a commercial ESG, and its failure to be included does not mean that the enterprise's ESG performance could be better. Using a less recognized indicator as the primary research object, the author should do more robustness.

The article could be easier to read, and the author is suggested to polish the language. Much of the paper should be reorganized.

IJERPH is a highly influential international journal. When I read it, I referred to the literature you mentioned. Too many are Chinese literature, which may affect the judgment of readers. The document format and quotation format are incorrect, which should not be.

Reference

[1] GOODMAN-BACON A. Difference-in-differences with variation in treatment timing [J]. Journal of Econometrics, 2021, 225(2): 254-77.

[2] BAKER A C, LARCKER D F, WANG C C Y. How much should we trust staggered difference-in-differences estimates? [J]. Journal of Financial Economics, 2022, 144(2): 370-95.

Round 2

Reviewer 2 Report

  • The authors have answered my question well.

Author Response

Thank you for your recognition of our manuscript!

Reviewer 4 Report

I want to thank the authors for addressing my previous comments. Your manuscripts is much improved. However, I still have a few comments:

Picture 1 needs to be further clarified. Now it looks like a screenshot. In the parallel trend test, why is there only one period and three periods in advance, but no coefficient for the second period (pre2)? It is hard to be convinced from the current inspection results.

In line “We next applied the triple difference method (DDD).” But where is the result?

The coefficients in Table 6 and 7 cannot be directly compared, and additional tests are required.

I think Part VII is unnecessary and can be incorporated into Part VI.
